# Effect of Grazing on the Welfare of Dairy Cows Raised Under Different Housing Conditions in Compost Barns

**DOI:** 10.3390/ani14233350

**Published:** 2024-11-21

**Authors:** Beatriz Danieli, Maksuel Gatto de Vitt, Ana Luiza Bachmann Schogor, Maria Luísa Appendino Nunes Zotti, Patrícia Ferreira Ponciano Ferraz, Aline Zampar

**Affiliations:** 1Graduate Program in Animal Science, Santa Catarina State University (UDESC), Chapecó 89815-630, Brazil; mak-witt@hotmail.com (M.G.d.V.); ana.schogor@udesc.br (A.L.B.S.); maria.anunes@udesc.br (M.L.A.N.Z.); aline.zampar@udesc.br (A.Z.); 2Department of Agricultural Engineering, Federal University of Lavras (UFLA), Lavras 37200-000, Brazil; patricia.ponciano@ufla.br

**Keywords:** animal behavior, ruminant production, animal production system

## Abstract

This study investigates whether the welfare of dairy cows housed in compost-bedded pack barns is enhanced by access to pasture. Our objective was to assess the welfare and daytime behavior of dairy cows in various compost-bedded pack barns across Brazil. Regardless of barn characteristics or pasture access, all farms were classified as “improved” according to the Welfare Quality^®^ protocol. However, dairy cows with access to pasture demonstrated higher scores in the “appropriate behavior” principle. Notably, during the colder months, there was a significant increase in the incidence of diarrhea among cows, which is attributed to seasonal dietary variations. In conclusion, the compost-bedded pack barn systems in Brazil provided favorable welfare and daytime behavioral conditions for the cows, and access to pasture further enhanced the welfare of animals in part-time housing.

## 1. Introduction

There exists an intrinsic relationship between welfare, health, and sustainability on dairy farms. To enhance the production system, it is essential to encourage the development and dissemination of knowledge in this area. The Welfare Quality^®^ (WQ^®^) protocol brings together the five freedoms proposed by the Farm Animal Welfare Council into four principles, which can be used for animal welfare assessments [1]. In the WQ^®^ protocol, scores are assigned for the animal welfare principles, defined by good nutrition, good housing, good health, and appropriate behavior. The score of each principle serves to establish strategies, regulating whether the production system is acceptable or not, considering respect for the physical and mental state of the animals, which implies good health and a feeling of animal welfare [1]. The overall WQ^®^ score is used to classify the welfare of dairy cows, such that properties and systems can be compared equally [1].

The WQ^®^ protocol has been applied to understand whether husbandry systems affect the welfare of dairy cows in Brazil [2] and worldwide [3,4,5,6,7,8]. Wagner et al. [6] and Burow et al. [9] studied WQ^®^ in herds kept in pasture, Molina et al. [10] analyzed dairy farms in Spain, Coignard et al. [3] and Des Roches [11] evaluated French herds, and Popescu et al. [4] evaluated housing systems (loose housing vs. tie-stall). In Brazil, the WQ^®^ protocol was used on pasture, compost-bedded pack barns (CBPs), and free stall farms, and the results showed that the pasture-based system scored better in most measures. Among confined systems, CBPs showed advantages in the principle of good housing [2].

Until now, we know that the cows housed in CBPs have a lower prevalence of lameness and better leg and foot health when compared to cows housed in other systems [12], especially when the system allows access to pasture [13]. Nevertheless, cows may be dirtier due to the physical condition of the litter, which occurs more often in the rainy season [12,14]. The overall WQ^®^ score of dairy cows is better when the housing system allows access to pasture [8]. In Brazil, CBP systems with access to pasture or zero access to pasture were identified [13,15]. Although the CBP system receives a good score on the principle of good housing [2], we do not know if this relationship applies to all types of CBPs in Brazil.

It is thought that the type of housing, management aspects in CBPs, or access to pasture can have strong effects on the animal welfare of dairy cattle. Nevertheless, there is no information regarding the general welfare conditions of dairy cows in CBPs that allow access to pasture. The objective of this work was to investigate and classify the welfare and behavior of dairy cows in three housing conditions in CBPs during both cold and hot seasons in southern Brazil.

## 2. Materials and Methods

### 2.1. Farms

This study was carried out in the cold season (August to October) of 2018 and the hot season (January to March) of 2019 on nine dairy farms with a CBP housing system in Santa Catarina State, Brazil. The farms were visited on four consecutive days in each season. The selection of farms was determined by the structural characteristics indicated in Radavelli’s study [15].

The study by Radavelli et al. [15] categorized CBPs in Brazil into three groups with similar characteristics, which informed the selection of the nine farms for this study. Each group adhered to the classifications established by Radavelli et al. [15]:(a)The CONV group consisted of large conventional CBPs that were permanently used, exclusively featuring new and larger barns (bedded pack areas ranging from 1301 to 2800 m^2^), similar to the American models outlined by Janni et al. [16].(b)The ADAP group included conventional or adapted CBPs that were also permanently used. These barns could be constructed according to the layout defined by Janni et al. [16], provided that they had bedded pack areas between 285 and 1300 m^2^ or were adapted from other rural facilities with varying bedded pack sizes.(c)The PART group comprised CBPs with partial use and access to grazing areas. Regardless of size, these barns were utilized only during the hottest parts of the day or during the rainy season, lacking mechanical ventilation and tilling of the bedded pack area.

The CONV group’s stables in this study had bedding areas ranging from 900 to 1500 m^2^, equipped with mechanical ventilation and sprinklers in the waiting area or along the track. These stables were constructed from precast concrete and featured tin roofs. Two of the farms milked cows twice a day, while one farm employed a robotic milking system.

The ADAP group’s stables had bedding areas ranging from 975 to 1470 m^2^. One barn was specifically designed for dairy cows, while the other two were adaptations of meat poultry buildings, lacking a consistent construction standard. All three ADAP stables were equipped with mechanical ventilation, though only one included sprinklers in the feeding lane. Of the three farms, one milked cows twice a day, while the other two milked three times a day.

The PART group’s stables had bedding areas ranging from 220 to 480 m^2^ and provided cows with access to grazing areas. Farms one and two utilized the stable primarily for supplementation (silage and concentrate), while farm three allowed free access to the stable during the hottest parts of the day, particularly in the afternoon, to shield the cows from direct sunlight. Farms one, two, and three provided grazing times of 22, 14, and 17 h, respectively. These stables lacked mechanical ventilation, and tilling of the bedding was not commonly practiced. Milking frequency for all farms was twice a day. The characteristics of the farms, stables, and the daily management of each farm are presented in Table 1.

### 2.2. Microclimatic Characteristics

The external microclimate of each stable was measured over four consecutive days in each season. The microclimate was characterized by dry bulb temperature (TBS, °C), black globe temperature (BGT, °C), and relative humidity (RH, %). TBS and RH were recorded using a data logger (Hobo, U12-013, Onset’s, MA, USA). The BGT was measured using an external temperature sensor (Hobo, TMC6-HD, Onset’s, MA, USA) encased in a hollow polyethylene sphere painted matte black (BGT sensor). The data logger, connected to the external sensor, was positioned 1.5 m above the ground and approximately 10 m from the stable, inside a meteorological shelter, except for the BGT sensor. The temperature and humidity index (THI) was calculated according to [17], and the black globe temperature index (BGTI) was calculated according to [18] (Table 2).

### 2.3. Evaluation of Animal Welfare

The reference material for assessing cow welfare in this study was the WQ^®^ protocol [1], which includes 29 measurements collected on the farm, grouped into 11 criteria and then finally 4 principles. On each farm, two trained evaluators assessed the cows. Data collection was conducted in accordance with the protocol’s guidelines [1]. Most information was recorded directly at both the animal and herd levels. Some measures related to agricultural resources and management were assessed based on farmers’ statements.

All cows in CBPs were evaluated; therefore, the number of cows is detailed in Table 1. All evaluations were conducted under consistent management conditions. Clinical scores—such as body condition score, body cleanliness, diarrhea and integumentary changes, and nasal, ocular, and vulvar secretions—were recorded during feeding, following the morning milking. To detect lameness, cows were observed as they exited the milking parlor. The assessment of the avoidance distance began 30 min after the cows arrived at the feed/neck rack, adhering to the guidelines of the assessment protocol for recording measurements.

Behavioral observations—such as the time taken to lie down, collisions with equipment while lying down, animals lying partially or completely outside the resting area, coughing, and expressions of agonistic behaviors—were conducted through visual observation by two trained observers. The observation period lasted 120 min and was carried out in one to three segments of the facility, depending on the number of lactating cows in the barn [1]. Observers maintained a sufficient distance to monitor behavioral activity influencing the animals’ natural behavior. Qualitative behavior was identified in the same location and at the same time as the behavioral observations.

Furthermore, as part of the protocol implementation, resources such as water point measurements were evaluated. Following assessments related to the animals and resources, technical information was gathered on access to pasture, dehorning or tail docking procedures, and health metrics (including milk somatic cell count, mortality, dystocia, and downer cows). The application of the WQ^®^ protocol in the PART group was conducted exclusively inside the stable and this methodological definition did not influence the data, as the cows’ duration of stay exceeded 120 min.

### 2.4. Daytime Behavior Assessment

On the same properties, two mornings were used for the daytime behavior protocol. One of the trained evaluators observed and recorded the daytime behavioral patterns of the cows. The monitoring began approximately two hours after the morning milking, equivalent to the evaluation period between 8:00 and 12:00 h (totaling 4 h assessed). The behaviors were recorded every 15 min using the instant scan method for a period of 4 h per day [19]. During each period, the following behavioral states were recorded: food intake, water intake, lying in a bedded pack, idle standing in a bedded pack, and idle standing on the feeding track. All lactating cows present in the barn of each farm were included (see Table 1).

To record these behavioral states, the following characterizations were considered: (a) food intake: when the cow lowers its head to ingest or not eat the food in the feeder; (b) water intake: when the cow stands in front of the water trough, whether drinking the water or not; (c) lying in a bedded pack: when the cow has more than 50% of its body in contact with the bedded pack, that is, lying on the CBP bedding, chewing its cud or not; (d) idle standing in bedded pack: when the cow with more than 50% of its body standing on the bedding without being involved in any other apparent activity; (e) idle standing on the feeding track: when the cow is standing with more than 50% of the body in the feeding area without being involved in any other apparent activity.

The total time of CBP use on the farms in the PART group was variable. We identified that in two farms in the PART group, the cows spent less than four hours, compromising the execution of the behavioral assessment. Therefore, the behavioral assessment of the PART group was conducted on only one of the farms, meaning that two farms were disregarded.

### 2.5. Statistical Analysis

The measurements collected and used to classify welfare quality were processed using WQ^®^ software (https://www1.clermont.inra.fr/wq/), generating criteria and principles. Data were expressed as the number of affected animals in relation to the total number of animals evaluated on each farm. In each group, descriptive statistical indicators (median and standard deviation) were determined for 29 measurements. The criteria were: absence of prolonged hunger, absence of prolonged thirst, comfort around rest, ease of movement, absence of injury, absence of disease, absence of pain induced by management procedures, expression of social behaviors, expression of other behaviors (indicates access to pasture), good relationship with humans, and a positive emotional state. In each group, descriptive statistical indicators (median and standard deviation) were determined for principles (good nutrition, good housing, good health, and appropriate behavior). Finally, farms were classified into a welfare category: not classified, acceptable, improved, or excellent [1].

For the statistical analysis of the measurements, criteria, and principles derived from the WQ^®^, a randomized block design was adopted in a 2 × 3 factorial scheme (two climatic seasons and three groups). The residual normality test was performed, and the statistical analysis was performed using SAS^®^ 9.4 software. When the normal distribution of the residual was not met, the variable was transformed. The means of the groups, the climatic seasons, and the interactions between groups and seasons were compared using the Tukey test at 5% probability. The daytime behavior of cows was investigated by the frequency of each behavior within each group and each season.

The principles of absence of prolonged thirst, ease of movement, and expression of other behaviors, as well as the measures of the percentage of collisions while lying down, the percentage of animals lying outside the rest area, and respiratory difficulties, could not be analyzed statistically due to insufficient variability for an analysis of variance.

## 3. Results

### 3.1. Animal Welfare

Table 3 presents the WQ^®^ protocol measurements and their median scores for the CONV, ADAP, and PART groups in two seasons of the year (cold and hot). The scores for the 11 criteria and four principles in the three housing systems (CONV, ADAP, and PART) and in the two seasons are presented in Table 4. The interaction between groups and seasons was insignificant for all measures, criteria, and principles. Among the measurements, only the duration of lying down movements and the percentage of cows with diarrhea showed statistically significant differences between seasons. In the cold season, cows needed significantly more time to lie down compared to the hot season. We also observed that the percentage of cows with diarrhea was significantly higher in cold weather than in hot weather. Cows in the PART group had a significantly higher frequency of butts/cow/h than the other groups.

Ease of movement, absence of prolonged thirst, and expression of other behaviors could not be analyzed statistically. None of the 11 criteria differed between groups or seasons. On the other hand, only the appropriate behavior principle varied between groups, being higher in the PART group than in the CONV and ADAP groups. Finally, regardless of the season, all the farms were classified as improved. No farms were classified as excellent, acceptable, or unclassified.

### 3.2. Behavior of Cows Housed in CB

Table 5 presents the average frequency of behaviors observed in the different CBP groups over two seasons. Among the behaviors monitored in the CBP groups, the one most frequently performed by cows was the act of lying in a bedded pack (Table 5). The cows housed in the CONV or ADAP systems maintained practically the same proportion of activities, regardless of the season. This result suggests that both production systems led to similar daily behaviors in the animals throughout the year. In contrast, the cows in the PART group were predominantly lying down, regardless of the season, as the CBP was designated for resting.

## 4. Discussion

### 4.1. Assessment of Welfare Measures Using WQ^®^

Regardless of group or season, the incidence of lean cows was low. Blanco-Penedo et al. [20] reported that the body condition scores of cows housed in CBPs were adequate for 92% of the cows, which aligns with the findings of this study. Farmers often perceive grazing as an economical method to reduce feed costs; however, for cows with high milk production, it is essential that grazing meets their metabolic needs to prevent a decline in body condition [8]. Grazing dairy cows are at a greater risk of inadequate nutrient intake, which may result in lower body condition scores compared to cattle kept exclusively indoors [8,9].

The farm must provide a linear drinking trough with a minimum length of 10 to 15 cm per animal and at least two water points per stable to prevent limitations on water consumption due to dominance behavior among the cows [10]. Approximately 65% of the farms did not meet the minimum linear drinker trough requirements per animal.

Additionally, some farms failed to properly manage the cleaning of the drinking trough, which may have resulted in reduced water intake. While all farms in the PART group had water sources in pastures, the number and size of drinking troughs were insufficient, necessitating that cows drink water in the barns. The primary barrier for farmers in installing waterers in pastures was cost, as many grazing areas were located considerable distances away, making a water piping system potentially expensive [8].

The time required for a cow to lie down should be 5.20 s or less [1]. The time observed in this study was similar to the times of 3.6 and 3.3 s recorded for CBP and grazing systems, respectively [2]. The longest time taken for cows to lie down during the cold season did not exceed the appropriate limit. The increased time required for cows to lie down in the cold season may be attributed to the litter surface being wetter or less soft during this season, which impairs the cow’s movement [4].

A CBP is a stable that is free of stalls, partitions, or moorings, providing cows with access to litter and the opportunity to exercise together [21]. The ADAP and PART groups had pillars in the resting area; however, no collisions were observed while the cows were lying down, similar to findings in the CBP and pasture-based systems evaluated in the State of Santa Catarina [2]. Any obstacle, such as a pillar or wall, poses a risk of serious injury if struck by cows [19]. This outcome is positively associated with cow productivity, health, and welfare [22].

The cleanliness of cows may vary depending on farm management practices [10,21], as well as the specific body parts observed [10,11]. In the present study, the percentage of cows with dirty legs was notably high; however, these values were deemed acceptable [11,21]. Blanco-Penedo et al. [20] found that 62% of cows housed in CBPs had dirt on their legs. Animal density and litter moisture are predictors of cleanliness scores (for legs, roofs, udders, and flanks), with higher litter moisture or animal density correlating with increased dirtiness [14]. Maintaining the cleanliness of cows is particularly challenging during rainy and humid seasons.

We observed a low prevalence of lameness within the herd, which aligns with findings from other studies on CBPs, where lameness was reported in approximately 4.4% to 9.1% of the herd [12,23]. Blanco-Penedo et al. [20] identified mild and severe lameness in 22% of cows sampled in CBP contexts. The results of this study suggest that the quality of the litter contributes to the low incidence of lameness, as Alsaaod et al. [24] also indicated. This system allows hooves to sink into the bedding, providing better traction and facilitating movement.

Skin changes may be associated with the incidence of lameness [3,25]. In this study, no significant influence of stable type or season on integument condition was observed. However, access to grazing has been reported to benefit foot health [6,9]. Blanco-Penedo et al. [20] found that approximately 52% of cows housed in CBPs exhibited areas of hair loss, which is higher than the values reported in the present survey.

The WQ^®^ system facilitates a comprehensive assessment of animal health by integrating various measures that may adversely affect dairy cow welfare [1]. The overall health assessment of the study herds was consistent with results reported for French dairy herds (the system was not specified by the authors) [3] and European herds maintained on pasture [6].

Among the measures included in the health assessment, the occurrence of mastitis is particularly significant due to its association with pain and its direct impact on milk quality. In CBPs, this factor receives special attention, as organic material in resting areas may predispose cows to mastitis. According to Fávero et al. [14], excessive moisture in the bedding predicts the growth and proliferation of microorganisms, increasing cows’ exposure to these pathogens. This issue can be exacerbated in wetter climatic conditions [12]; however, it was not observed in this study.

Eye discharge levels were low across the groups and seasons evaluated. Blanco-Penedo et al. [20] noted that eye discharge was common in cows housed in CBPs. It is important to emphasize the implications of eye discharge in stables, as it may lead to an increase in the incidence of flies and higher emissions of ammonia or dust [6].

Respiratory difficulties were more prevalent in the PART group, particularly during the hot season, suggesting potential challenges associated with the unstable climatic conditions. It is also noteworthy that the CBP farms did not provide mechanical fans or nebulizers to enhance thermal comfort.

The percentage of cows experiencing diarrhea was significantly higher in the cold season compared to the hot season. In contrast, Burow et al. [9] and Wagner et al. [6] reported the opposite trend for grazing herds. One possible explanation for the increased incidence of diarrhea in the cold season is a nutritional imbalance, especially in the PART group, as it is vulnerable to rapidly changing weather and feed conditions. Sudden dietary changes, along with the ingestion of large quantities of grass, can lead to considerably loose manure in grazing cows [8].

Popescu et al. [4] reported that group-housed cows exhibited more aggressive behavior than tethered cows; however, they also noted that a lack of interaction between animals could be detrimental to welfare. In the social behavior observed among the cows, several individuals were seen running in the litter and displaying estrous behaviors, such as mounting other animals. This estrous behavior facilitated social interactions, including head-butting, stalking, and fighting, consistent with findings from Grimard et al. [26]. Endres and Barberg [19] also reported aggressive events among dairy cows housed in CBPs, documenting an average of 0.94 chasing-up events, 0.94 chases, and 1.4 head butts per hour.

De Vries et al. [25] suggest that the frequency of displacements—interactions involving physical contact where one animal pushes or strikes another, causing the latter to relinquish its position—is primarily influenced by competition for limited resources. In this study, most agonistic interactions occurred near the drinking fountains and the feeding aisle. Similarly, Miller and Wood-Gush [27] found that the majority of agonistic interactions took place in the feeding aisle.

Cows in this study displayed greater resistance to human contact compared to those in other studies [4]. However, this relationship does not imply that the system is inadequate. Rather, it suggests the presence of issues in animal–human interactions that may not have been significantly influenced by management practices.

### 4.2. Criteria Scores and Principles of WQ^®^

Measurements were organized into criteria and subsequently grouped into principles to facilitate comparison between groups and seasons. Only the proper behavior criterion varied among the groups, as access to grazing is a significant factor in the scoring of “proper behavior” within the WQ^®^ protocol [1]. Consequently, the farms in the PART group received higher ratings. In contrast, the CONV and ADAP groups received lower scores due to their lack of grazing access.

Water supply is a critical factor affecting the welfare and productivity of dairy cows, and farmers should be mindful of this issue. However, the absence of prolonged thirst did not receive the highest score in any of the groups and seasons. This indicates significant management shortcomings, as prolonged thirst has more detrimental effects on animal welfare than prolonged hunger, particularly for dairy cows, which have increased water consumption during lactation [4].

One of the criteria included in the principle of “good housing” is the comfort around rest, which received scores exceeding those reported in the literature for cows housed in tie-stalls, loose housing, or grazing systems [4,6]. This finding suggests that cattle housing in CBPs promotes better comfort conditions, irrespective of the season. All treatments achieved maximum scores (100) for ease of movement, as the CBP system is free from tethers.

Comparisons of the measures constituting the principle of “good health” are challenging, as the issues primarily relate to the management practices of the respective regions. Consequently, no criteria were significantly influenced by group or season (Table 4). It is noteworthy that, for the good relationship with man criterion, only management practices related to dehorning were considered. This is because tail anchoring is not a common practice in Brazil, whereas dehorning is prevalent across all farms in our study, which accounts for the non-maximum scores observed. The dehorning procedure was performed using thermocautery during the lactation phase on all farms. Some farms administered the procedure with anesthetic, a practice more frequently observed in the CONV and ADAP groups, which received higher scores.

Proper behavior in the PART group was significantly greater than in the other groups, and the CONV and ADAP groups did not show any differences in this parameter. Access to grazing did not confer any advantages in terms of measurements or criterion scores. This suggests that, regardless of group, farmers were generally motivated to enhance farm management or stable conditions, potentially accounting for the similar results. Pasture-based systems are perceived to provide greater behavioral freedom compared to those with continuous housing [2,6,8].

### 4.3. General Evaluation of WQ^®^

Regardless of the season, all farms were classified as improved, having received scores greater than 15 for all principles and exceeding 50 for at least two [1]. It is essential to highlight the positive impact of grazing on these scores. The results indicate that all farms achieved favorable animal welfare, which was enhanced by the type of housing and access to grazing. Although all farms were classified as improved, grazing demonstrated the potential to further enhance the animal welfare of dairy cows throughout the year. However, this beneficial effect can only be assured if the other principles adequately meet the animals’ needs. Furthermore, it is known that CBPs were neither designed nor their management practices defined for partial use. Thus, although we found positive results, this may represent a limitation in applying these findings, as no similar cases have been reported in the literature for other regions in Brazil or countries that use CBPs in this way.

When comparing cows housed in free stall systems with those in CBPs and on pasture, the latter two systems exhibited significant advantages across all animal welfare principle scores according to the WQ^®^ protocol [2]. Specifically, when comparing farms with pasture access to those without (zero pasture), approximately 81% of farms with pasture access were classified as improved, while around 71% of farms without pasture access were deemed acceptable [8].

### 4.4. Daytime Behavior

In this study, the cows spent the majority of their time lying down rather than engaging in other activities. It is important to note that dairy cows typically lie down for approximately 9 to 12 h per day [19,22]. Additionally, this behavior can be influenced by factors such as the stage of lactation, time of day [19], or feeding frequency [22]. Using a similar behavioral methodology, Endres and Barberg [19] reported that an average of 43.3 ± 28.5% of cows were lying down at any given time.

Munksgaard et al. [28] found that lying down took precedence over other activities, such as eating or social interaction. Their observations indicated that on farms with three milking sessions per day, cows prioritized feeding and seeking out their resting areas, resulting in limited time for other behaviors, such as standing idly in the passageway or in their resting areas.

Cows in the PART group spent more time lying down compared to those in other groups, as this system was specifically designed to facilitate resting. Our observations indicated that cows in the PART group laid down approximately 19% more during the cold season than in the hot season, despite experiencing similar occupancy levels in both seasons. It is important to note that at the farms in the PART group, where behavior was monitored, drinking fountains were located at one end of the stable, which hindered the cows’ access. During the cold season, access to the fountains was challenging because the cows were evenly distributed throughout the bedding area. In the hot season, the cows positioned near the drinking fountains were often unable to lie down comfortably, as they needed to make room for others to access the water.

During the hot season, cows tended to seek the water cooler more frequently. In the present study, the overall THI remained above 72 throughout the summer; however, the cows spent less time lying down. Endres and Barberg [19] similarly noted that cows decreased their lying time as THI increased. Allen et al. [29] also observed a higher proportion of standing cows when exposed to THI values between 80 and 89. This behavior of remaining standing longer under such thermal conditions facilitates heat dissipation into the environment. Nevertheless, Endres and Barberg [19] suggest that these behavioral changes may indicate restlessness, stress, or an increased need to walk to the water cooler during warmer periods. Access to drinking fountains is crucial, as cows typically spend about 20 to 30 min per day drinking water, particularly in the summer [10].

In general, water intake was the least frequent activity observed across all evaluated groups and climatic seasons. This less frequent behavioral response occurred less often throughout the day compared to other behaviors. Access to water points typically followed activities such as lying in bedded packs or food intake.

When cows are engaged in idle standing, they prefer to do so in the resting area rather than at the feeding trough. Several factors contribute to this preference. The resting area usually features a soft, dry bedding surface that enables cows to stand and lie down comfortably [19], in contrast to the feeding trough, which does not provide the same level of comfort. The cows generally exhibited a daily behavior pattern characterized by feeding, followed by water intake and then seeking a comfortable place to lie down. The CBP system facilitates greater freedom of movement for the animals, making the ability to lie down and rest comfortably essential for their welfare [19].

## 5. Conclusions

The compost-bedded pack barn system provided adequate welfare conditions for cows housed in any season in Brazil. Access to grazing significantly contributed to the welfare of animals in partially used compost barns. Furthermore, some welfare measures and behaviors of dairy cows were influenced by the seasons of the year. The results indicated that the use of the compost-bedded pack barn was associated with improved welfare and the execution of appropriate daytime behaviors. However, there are few studies that have evaluated welfare using a protocol that allows for comparisons with other systems.

## Figures and Tables

**Table 1 animals-14-03350-t001:** Description of farms with a compost-bedded pack (CBP) housing system in Santa Catarina State, Brazil.

Group	Farm	Ceiling Height (m)	Drinker Length ^1^	Stocking Rate (m^2^/Cow)	Number of Lactating Cows
CONV	1	5.7	14.3	10.7	83.0
	2	6.5	19.5	24.5	61.0
	3	4.4	7.3	12.1	123.0
ADAP	1	5.1	9.1	16.8	70.0
	2	6.0	24.4	28.3	52.0
	3	3.0	6.1	16.4	73.0
PART	1	2.4	9.6	9.1	26.0
	2	3.2	4.5	3.4	32.0
	3	4.8	6.1	8	60.0

^1^ Length in centimeters of liners of drinkers per animal. CONV: large conventional CBP group. ADAP: group of conventional or adapted CBPs. PART: CBPs with partial use and access to grazing areas.

**Table 2 animals-14-03350-t002:** Comparison of the microclimates of the farms during the hot and cold seasons of the experiment.

Season	Variable	Mean	Maximum	Minimum
Cold	Dry bulb temperature (°C)	16.8	24.9	10.3
	Relative humidity of the air (%)	78.6	97.9	52.6
	Black globe temperature index (°C)	64.6	79.9	54.2
	Temperature and humidity index	61.4	72.0	51.1
Hot	Dry bulb temperature (°C)	24.5	31.5	19.8
	Relative humidity of the air (%)	80.9	97.1	60.9
	Black globe temperature index (°C)	75.3	87.1	67.8
	Temperature and humidity index	73.9	82.5	67.5

**Table 3 animals-14-03350-t003:** Scores of the Welfare Quality^®^ protocol measures in compost-bedded pack barn (CBP) systems during both cold and hot seasons.

WQ^®^ Measures	CONV	ADAP	PART	HOT	COLD	Overall *p*–Value	
	Median (SD) ^2^	Median (SD)	Median (SD)	Median (SD)	Median (SD)	Group	Season
Lean cows (%)	5.00 (6.92)	3.50 (3.43)	8.00 (3.42)	4.30 (3.77)	5.50 (5.86)	0.5528	0.4136
Time needed to lie down (s)	3.87 (0.57)	4.25 (0.55)	3.97 (0.56)	3.30 (0.45)	4.45 (0.43)	0.6966	0.0002
Collisions while lying down (%) ^1^	0.00 (0.00)	0.00 (0.00)	0.00 (0.00)	0.00 (0.00)	0.00 (0.00)	-	-
Animals lying outside the rest area (%) ^1^	0.00 (0.00)	0.00 (0.00)	0.00 (0.00)	0.00 (0.00)	0.00 (0.00)	-	-
Cows with dirty legs (%)	55.24 (20.80)	87.71 (27.48)	64.07 (27.48)	72.90 (26.56)	83.57 (20.80)	0.1239	0.6265
Cows with dirty udders (%)	15.49 (12.79)	22.60 (9.65)	10.09 (9.65)	24.10 (11.04)	7.37 (7.65)	0.4275	0.0833
Cows with dirty thighs and flanks (%)	35.01 (14.50)	37.71 (16.78)	16.94 (16.78)	39.50 (15.25)	28.08 (15.37)	0.1523	0.4648
Moderately lame (%)	6.98 (4.44)	9.61 (6.15)	7.99 (6.15)	5.90 (4.69)	10 (4.68)	0.9725	0.2383
Severely lame (%)	2.21 (1.23)	2.54 (2.79)	2.37 (2.79)	1.90 (1.11)	2.45 (2.31)	0.6619	0.4290
Mild integument alterations (%)	10.94 (6.10)	9.64 (3.47)	7.48 (3.47)	8.40 (3.57)	9.37 (6.90)	0.2535	0.3887
Severe integument alterations (%)	35.18 (10.03)	43.26 (10.76)	48.92 (10.76)	42.9 (8.29)	40.98 (11.24)	0.1679	0.7053
Coughing	0.02 (0.02)	0.02 (0.40)	0.05 (0.41)	0.10 (0.35)	0.02 (0.02)	0.2434	0.2432
Nasal discharge (%)	20.62 (7.67)	15.25 (4.68)	9.22 (4.68)	13.60 (7.39)	13.11 (7.15)	0.0518	0.6243
Eye discharge (%)	1.28 (0.79)	0.00 (1.21)	1.16 (1.21)	0.80 (1.12)	0.81 (3.00)	0.9055	0.4847
Respiratory difficulties ^1^	0.00 (1.43)	0.00 (6.56)	2.58 (6.56)	3.90 (5.39)	0.00 (0.00)	-	-
Diarrhea (%)	7.93 (3.37)	19.23 (8.02)	13.34 (8.02)	10.30 (4.97)	15.57 (6.65)	0.1066	0.0304
Vulvar discharge (%)	1.21 (1.14)	0.87 (1.26)	1.78 (1.26)	0.90 (1.16)	1.92 (1.18)	0.7776	0.2835
Mastitis (%)	0.00 (1.03)	0.00 (1.09)	1.00 (1.09)	0.00 (1.03)	0.00 (0.88)	0.2874	0.6545
Mortality (%)	3.61 (1.09)	5.76 (2.88)	3.27 (5.31)	3.33 (4.65)	3.33 (4.36)	0.4030	0.5240
Disability (%)	8.19 (10.95)	7.14 (2.62)	4.90 (2.62)	7.70 (7.53)	7.14 (7.08)	0.5486	0.9037
Downer cows (%)	3.33 (4.51)	3.84 (5.31)	3.27 (5.31)	3.30 (4.65)	3.33 (4.36)	0.8281	0.8909
Frequency of butts/cow/h	0.08 (0.02) b	0.06 (0.15) b	0.19 (0.15) a	0.10 (0.11)	0.08 (0.12)	0.0072	0.7863
Frequency of displacements/cow/h	0.09 (0.03)	0.12 (0.24)	0.14 (0.24)	0.10 (0.19)	0.12 (0.13)	0.2031	0.9733
Cows that can be touched (%)	33.70 (15.63)	28.76 (7.94)	30.47 (7.94)	32.30 (11.70)	28.76 (12.69)	0.8193	0.5618
Cows that can be touched up to 50 cm (%)	45.66 (9.70)	38.98 (11.77)	45.02 (11.77)	41.00 (7.38)	49.18 (11.15)	0.4697	0.0630
Cows that can be touched from 50 to 100 cm (%)	16.76 (5.65)	19.49 (5.85)	9.72 (5.85)	18.30 (4.48)	9.83 (9.13)	0.3855	0.4269
Cows that cannot be approached (%)	12.19 (8.66)	7.62 (6.42)	7.32 (6.42)	8.10 (10.00)	5.71 (8.74)	0.5067	0.5932

a,b Different letters within a row denote differences between groups (*p* < 0.05). CONV = large conventional CBP. ADAP = conventional or adapted CBP. PART = CBP with partial use and access to the grazing area. ^1^ The measures do not contain enough variability to be subjected to an analysis of variance. ^2^ Standard deviation. Obs. The interaction between group and season was insignificant for all traits.

**Table 4 animals-14-03350-t004:** Scores of the principles and criteria of the Welfare Quality^®^ protocol in the compost-bedded pack barn (CBP) systems during cold and hot seasons.

Principles and Criteria	CONV	ADAP	PART	HOT	COLD	Overall *p*–Value	
	Median (SD) ^2^	Median (SD)	Median (SD)	Median (SD)	Median (SD)	Group	Season
**Good nutrition**	76.37 (20.75)	84.30 (12.49)	55.45 (21.93)	79.12 (22.56)	73.70 (19.05)	0.4863	0.7224
APH	67.60 (19.65)	78.50 (13.96)	55.07 (14.55)	71.37 (15.41)	63.95 (18.26)	0.3589	0.5017
APT ^1^	100.00 (27.76)	100.00 (0.00)	60.00 (30.56)	100.00 (31.56)	100.00 (24.81)	-	-
**Good housing**	74.47 (4.91)	72.45 (2.35)	83.12 (9.61)	73.42 (8.74)	74.55 (6.55)	0.2822	0.7970
CAR	59.47 (7.79)	56.25 (3.74)	73.15 (15.24)	57.80 (13.86)	59.60 (10.38)	0.1252	0.9519
EM ^1^	100.00 (0.00)	100.00 (0.00)	100.00 (0.00)	100.00 (0.00)	100.00 (0.00)	-	-
**Good health**	28.92 (10.80)	32.35 (3.32)	32.40 (4.86)	32.72 (6.96)	30.75 (7.39)	0.8051	0.9494
AI	52.32 (10.12)	50.70 (7.84)	49.30 (15.01)	51.77 (10.55)	50.10 (12.04)	0.7227	0.5422
AD	25.32 (12.77)	24.85 (3.70)	30.37 (9.78)	27.87 (10.43)	27.40 (9.06)	0.7525	0.9437
APIMP	52.00 (12.39)	52.00 (0.00)	28.00 (12.39)	52.00 (12.42)	52.00 (12.00)	0.1313	1.0000
**Proper behavior**	24.15 (2.50) b	23.50 (4.96) b	41.02 (6.58) a	26.87 (9.58)	25.80 (11.26)	0.0139	0.4729
ESB	92.50 (5.04)	91.35 (3.77)	88.52 (14.81)	90.22 (11.55)	91.35 (9.47)	0.2217	0.8021
EOB ^1^	0.00 (0.00)	0.00 (0.00)	100.00 (0.00)	0.00 (51.75)	0.00 (50.00)	-	-
GRM	46.25 (11.82)	49.55 (15.09)	52.60 (8.37)	49.50 (10.57)	53.70 (13.03)	0.5144	0.5855
PES	43.35 (6.67)	33.70 (10.48)	25.12 (13.77)	32.02 (12.69)	38.15 (12.07)	0.0842	0.4368

a,b Different letters within a row denote differences between CBP systems (*p* < 0.05). ^1^ For the APT, EM, and EOB principles, there is not enough variability to subject them to an analysis of variance. ^2^ Standard deviation. CONV = large conventional CBP. ADAP = conventional or adapted CBP. PART = CBP with partial use and access to the grazing area. APH = absence of prolonged hunger. APT = absence of prolonged thirst. CAR = comfort around rest. EM = ease of movement. AI = absence of injuries. AD = absence of disease. APIMP = good relationship with man. ESB = expression of social behaviors. EOB = expression of other behaviors. GRM = good man–animal relationship. PES = positive emotional state. Obs. The interaction between group and season was insignificant for all principles and criteria. In bold: principles defined by the Welfare Quality^®^ protocol.

**Table 5 animals-14-03350-t005:** Average frequency of daytime behaviors observed in cows housed in compost-bedded pack barns (CBPs) in two seasons.

Behavior	Cold			Hot		
	CONV (*n* = 3)	ADAP (*n* = 3)	PART (*n* = 1)	CONV (*n* = 3)	ADAP (*n* = 3)	PART (*n* = 1)
FI (%)	21.88 ± 3.0	26.06 ± 6.7	-	19.50 ± 1.4	28.43 ± 3.1	-
WI (%)	3.42 ± 0.9	2.25 ± 0.6	0.27	3.64 ± 2.3	3.68 ± 1.1	1.72
ISB (%)	18.18 ± 7.8	15.64 ± 11.6	12.01	18.84 ± 9.3	21.73 ± 6.8	35.49
ISFT (%)	10.93 ± 8.7	3.46 ± 1.3	11.27	9.26 ± 5.0	2.36 ± 1.0	5.37
LB (%)	45.38 ± 13.1	52.57 ± 12.0	76.61	48.76 ± 15.8	43.77 ± 9.7	57.40

CONV = large conventional CBP. ADAP = conventional or adapted CBP. PART = CBP with partial use and access to the grazing area. FI = food intake. WI = water intake. ISB = idle standing in bedded pack. ISFT = idle standing on the feeding track. LB = lying in bedded pack. Obs. The only CBP in the PART group that kept cows in housing for a minimum of 4 h did not provide feed in the bedding pack area; therefore, the behavior was not quantified.

## Data Availability

The data presented in this study are available on request from the corresponding author. The data are not publicly available due to personal reasons.

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
