# Peer review of "Effect of Grazing on the Welfare of Dairy Cows Raised Under Different Housing Conditions in Compost Barns"

_animals, 2024, doi:10.3390/ani14233350_

Round 1

Reviewer 1 Report

Comments and Suggestions for Authors

Dears authors,

The study evaluation time on each farm and season should be longer for better data sampling.

Reviewer 2 Report

Comments and Suggestions for Authors

General comments:

The paper is very interesting, but there are some concerns that need to be addressed before publication to improve its quality. The methodology and measurement processes are not sufficiently clear, and some parts of the results require modification. Clarifying these issues will strengthen the paper and improve its overall quality.

Abstract:

Line 22:  For the abbreviations used for different types of barns, I believe 'C' refers to Compost, not conventional. If that's not the case, then part-time barns should be abbreviated as 'BP.' Alternatively, consider adding another 'C' for conventional.

Line 24: It would be helpful to include a brief explanation of this protocol, perhaps in simple terms, clarifying what it measures in relation to welfare criteria. The same for the principles of behaviour—without some explanation, it becomes difficult for readers to follow the contents.

Line 26: Are you talking about all different type of barns here?

There is no explanation of the statistical analysis in this abstract.

Introduction:

This section is well-written; however, a summary of the current challenges related to barns and their impact on animal welfare in dairy cattle farming is missing. It can be added at the beginning of the introduction.

M&M:

Line 81-94: Do you think that different types of management, such as milking, could also influence the welfare protocol? Did you try to compare those within the same system?

Line 132-140: Have these behavioral observations measured individually or by group within a barn? If individually, how did you ensure accurate monitoring of each animal? Why didn’t you use technology such as sensors or video recording? This section needs to be clearer about how the monitoring and measurements were carried out.

Line 155: Feed intake of food intake? Please check across the text, and Tables.

 Line 183-188: Did you consider including any covariates in your model? For example, factors like the milking system, milk yield, time of day, location, and age could influence behaviour and welfare. These parameters might have an impact that should be taken into account.

Line 186: What do you mean by statistical analysis system? Do you refer to any particular software, model? Need to be clear.  

Line 196: What was the reason you couldn't analyse them? I assume it was due to insufficient data, as shown in the Table 3. is that correct?  It needs to be explained.

Table 3: In column 1, consider adding the scale in the bracket in front of each parameter, for instance (n), or (%). This will make it clearer.

Table 4: This table is confusing in its current format. At first glance, it appears that each principle has sub-classes; for example, I initially thought APH and APT were sub-classes of good nutrition, but that doesn't seem to be correct. Additionally, did you only conduct statistical analysis for proper behavior, or was it the only parameter that was significant? Is that why you've indicated differences between groups with different letters? There are also 16 parameters listed in the "Principle and Criteria" column, but only 15 values provided. It seems one of them is missing.

Table 5: Did you exclude the CBP group from the analysis for FI? If so, this should be mentioned, for instance in the table footnote.

Discussion:

The discussion section is well-written and effectively compares the results with other published research. However, there is a lack of discussion on the study's limitations, such as the use of subjective measurements, which could be more accurate if it replaced by technological tools. It would be helpful to include any experiences or recommendations for future research in this area, this can be added in conclusion section. Additionally, some references are repeated throughout the discussion; consider finding alternative references or presenting them differently.

Reviewer 3 Report

Comments and Suggestions for Authors

Title: Protocol Welfare Quality® in Compost Bedded Pack Barns

Authors: B Danieli et al.

Date: Sept 26 2024

General comments

I have carefully read the manuscript dealing with the assessment of the welfare of dairy cows in three different groups of compost barns using the Welfare Quality protocol. I consider the topic to be relevant and interesting for the dairy sector. However, the manuscript has several weaknesses that need to be taken into account and corrected if necessary.

My main concern is the statistical analysis. The statistical part is poor and insufficient, lacking a lot of information necessary for understanding and evaluating the results/manuscript. The authors need to explain which statistical method(s) and test(s) were used and the reason for choosing these methods and tests. Since this information is missing, I cannot assess whether the results are presented and explained correctly. In addition, there are several shortcomings and also errors in the tables and in the description of the results. I have made some comments and suggestions in the specific comments. The number of farms is also very small, especially as some measurements were not carried out on all farms. The manuscript will benefit from an English checking.

Specific comments

Overall: Compost barn written with capital letters in many places in the manuscript. Please correct.

Title: Title is not informative enough. I also suggest to omit “Welfare Quality® protocol” from the title as also other measurements were made (daytime behavioral patterns are not according to the protocol).

l. 18: I suggest to omit the abbreviation in simple summary or alternatively explain it.

l. 19: It is not clear what do you mean by the word parameters? Also in line 63.

l. 72-87: The difference between a) and b) is not clear. According to the description in the first paragraph, the differences is in size, but in the next to paragraphs, the size of beds is almost the same. Explain it better – the reader must know the difference without looking at the reference ([15], [16]). What does the term "bed" refer to?

Table 1:

-       Remove (n=9) from the title – it is clear that there are nine farms.

-       Why is compost barn written with capital letters?

-       What is linear centimetre?

-       Remove semicolon ( ; ) after 2.45.

-       Unify the number of decimals within each column. I think one decimal would be enough.

-       Put m²/cow in brackets.

-       What is area 1 and 2?

-       Uppercase numbers 2-4 are not needed. Simply explain the abbreviation in the footnotes.

l. 104: Wet bulb globe temperature written with capital letters. Is this OK?

l. 108-109: Use space in front of unit, e.g. 1.5 m. In manuscript some measurement units are written as abbreviations and other with the whole word. Please unify according to the journal rules.

l. 110: I suggest to add “according to” after was calculated.

l. 110: Last sentence is incomplete.

Table 2:

-       I suggest to change the title as it is stretched out and clumsy written. Omit the word each. E.g. Microclimatic conditions on farms during the hot and cold season in Brazil (n=9).

-       Why if wet bulb globe temperature written with capital letters?

-       Why is the degree sign underlined in this table?

-       The word Cold is bold and Hot is not bold. Is it OK?

-       I think one decimal place is enough.

l. 124: Do you mean in each season instead of from each season?

l. 130-131: Why is “the assessment followed the instructions of the assessment protocol for recording the measurement” repeated here. This probably applies to many other measurements and observations made in your study, right?

l. 133: Is listless behaviour explained in the WQ protocol? If not, please add what did you observe/measure?

l. 137: What do you mean by “the same areas adopted for behavioral observations”?

l. 139: Change application time to assessment time.

l. 139-140: Sentence not clear.

l. 144: I suggest to remove “conditions of”.

l. 147-148: Sentence not clear. Which factor? Influence of which data?

l. 150-151: What do you mean by alternate days?

l. 152: You say at the same time, but then you give a period of 4 hours – at 8 or at 12 p.m. is not at the same time.

l. 172: What is key score? Do you mean principle score?

L. 173-179: See my comment on abbreviations below (referring to l. 193-195).

l. 180: Replace accommodation by housing. Also in some other places in manuscript.

l. 183-189: Statistical analysis is not sufficiently explained. What was the distribution of variables? What model did you apply?

Table 3:

-       Title is not OK. Please rephrase – e.g. The effect of group, time and their interaction on….

-       As already said, it is not clear which statistical method was used to obtained p-values. This information must be provided in statistical part of Material and methods.

-       Use G×S instead of I. A tip: The interaction G×S is insignificant in all cases. You can include this information in the footnote instead of writing NS in the whole column (the last column could then be omitted).

-       The presentation of the results is inappropriate. As the interaction between the effects G×S is not significant, the results should be presented separately by group (CBA, CBL, CBP) and season (C, H) and not as an interaction (CBA-C, CBA-H, CBL-C, CBL-H, CBP-C, CBP-H). The presentation used is only appropriate if the interaction is significant. Consequently, the ab values used to indicate significant differences between treatments are also not appropriate. Furthermore, ab must be explained in the footnotes.

-       Uppercase numbers are not needed. Simply explain the abbreviation in the footnotes.

-       Units should be placed with the variables in the table not in footnotes.

-       Why did you decide to show average (maximum-minimum) values for the variables in this table? If the variables are normally distributed you should use mean ± standard deviation. If they are not normally distributed (which is more likely), I suggest to provide median (first and third quartile); mean is not appropriate in this case.

Table 4: See comments for Table 3; the most of them are valid also for this table.

-       Why there is no measure of variability?

l. 192: There are not average scores for each farm, but for each of the three groups of farms.

l. 193-195: There are strict general rulers for the use of abbreviations. If you decide to use abbreviations in the text, please respect that the abbreviation should be explained when first used in the text. In subsequent occurrences, use only the abbreviation itself. Please respect this throughout the text. (However, abbreviation must additionally be explained in the abstract and in the tables/figures).

l. 195: This information is more suitable for the MM section. The reason needs to be provided why they were not statistically analysed.

L. 195-196: It is written that some variables cannot be statistically analysed, but then the statistical results (p-values) are provided for these variables in the table. Suggestion: For the variables that were not analysed, you can only give descriptive statistics (e.g. mean, median, quartiles), but for the p-values you put a sign like / or - . The result for the frequency of butt/cow/h is particularly strange. In Table 1, statistically significant differences are marked, although this behaviour never occurred in any of the groups. Please check the results again carefully.

l. 201-202: Again, statistically significant difference is indicated between groups for the frequency of butts. I understand, that this is a mistake, but please correct.

l. 204-206: If four of 11 criteria were not statistically analysed, then it is not possible to provide results for them (p-values, differences denoted by ab).

l. 206: What is eras?

l. 228: Delete and after CB.

l. 231-232: It is not clear how did you test this.

l. 227-234: No need for two paragraphs. Again, inconsistent use of abbreviations. I suggest not to use abbreviation for variables (expect maybe in tables) as there will be a great confusion as the abbreviations are used to groups and seasons too.

l. 380: The subtitle is same as in 4.3.

l. 423: Did you mean effects on behaviour?

l. 423-424: Sentence not clear.

Due to my many concerns on the statistics, I did not review the discussion part at this point.

All the best!

Comments on the Quality of English Language

Poor, simplified language/terminology. The manuscript contains a lot of empty text, ballast. It appears that the text was originally written in the native language and then translated directly into English, so it needs extensive editing.

Round 2

Reviewer 1 Report

Comments and Suggestions for Authors

Dears authors, 

According to the response in your cover letter, there is no way to change the methodology, as the study has already been done. Therefore, the considerations of the initial opinion continue.

Kind Regards.

Reviewer 2 Report

Comments and Suggestions for Authors

Dear authors,

Thank you for the responses to my comments on your manuscript. I appreciate the effort you put into addressing each point, and I am satisfied with the revisions. The changes you made have strengthened the manuscript, and I am happy with the revised version.

Best regards,

Reviewer